# Combined Effect of Leisure-Time Physical Activity and Sedentary Behavior on Abdominal Obesity in ELSA-Brasil Participants

**DOI:** 10.3390/ijerph20156501

**Published:** 2023-08-02

**Authors:** Maiara do Espírito Santo Cerqueira de Araújo, Maria da Conceição Chagas de Almeida, Sheila Maria Alvim Matos, Maria de Jesus Mendes da Fonseca, Cristiano Penas Seara Pitanga, Francisco José Gondim Pitanga

**Affiliations:** 1Postgraduate Program in Rehabilitation Sciences, Multidisciplinary Institute for Rehabilitation and Health, Federal University of Bahia (UFBA), Salvador 40110-170, BA, Brazil; pitanga@lognet.com.br; 2Gonçalo Moniz Institute, Oswaldo Cruz Foundation, Salvador 40296-710, BA, Brazil; conceicao.almeida@fiocruz.br; 3Institute of Collective Health, Federal University of Bahia, Salvador 40110-040, BA, Brazil; sheilaalvim@gmail.com; 4National School of Public Health, Oswaldo Cruz Foundation, Rio de Janeiro 21041-210, RJ, Brazil; mariafonseca818@gmail.com; 5Department of Physical Education, Catholic University of Salvador (UCSAL), Salvador 41740-090, BA, Brazil; cpspitanga@gmail.com

**Keywords:** physical activity, sedentary behavior, abdominal obesity, adults, health

## Abstract

Introduction: Abdominal obesity is a pattern of obesity that has been considered a public health problem. Physical activity is considered an important factor for the prevention of abdominal obesity. Increased time in sedentary behavior has been associated with negative health outcomes, including abdominal obesity. Objective: The aim of this study was to investigate which combination of leisure-time physical activity and sedentary behavior contributes most to the prevention of abdominal obesity in adults participating in ELSA-Brasil (Longitudinal Study of Adult Health). Methods: The study was cross-sectional and participants from the first follow-up of the ELSA-Brasil cohort (2012–2014) were analyzed. The independent variables were physical activity, assessed by IPAQ, and sedentary behavior, assessed by a standard questionnaire applied in ELSA-Brasil; the dependent variable was abdominal obesity, determined by waist circumference. The covariates analyzed were the following: age, education, binge drinking, smoking and menopause. The associations between the dependent variable and the independent variables were analyzed using logistic regression. The odds ratio with 95 CI% was estimated. Results: For men, the combinations were more significant when they were more physically active and spent less time on the sedentary behaviors analyzed, on both a weekday and a weekend day. For menopausal women, both younger and older, all associations of the combinations between sufficient leisure-time physical activity and little time spent in sedentary behaviors contributed to the prevention of abdominal obesity. In non-menopausal women, positive associations were observed in almost all combinations between leisure-time physical activity and sedentary behaviors, with some results that were not statistically significant among younger women. Conclusions: Our results showed that being sufficiently active and reducing the time spent in sedentary behavior was the combination that contributed the most to the prevention of abdominal obesity, both in men and women.

## 1. Introduction

The increasing prevalence of obesity in the world has been observed with great concern, as it is associated with an increased risk for the development of chronic diseases, cardiovascular diseases, and death [1]. Between 1975 and 2016, the prevalence of obesity increased significantly. In 1975, there were 100 million obese adults in the world [2]. In 2016, 39% of adults, aged 18 years and older, were overweight and 13% were obese [3]. The WHO estimate is that approximately 167 million people will have become overweight or obese by 2025 [4]. Between the years 2002 and 2019, the prevalence of obesity was observed to more than double for both men (9.6–22.8%) and women (14.5–30.2%) [5].

Obesity has concerned health organizations around the world. However, one type of obesity, abdominal obesity, has been considered a public health problem because it is associated with cardiometabolic diseases and death [6]. Abdominal obesity is independent of the presence of general obesity or being overweight [7], and is associated with poorer quality of life [6]. Due to its associations with negative health outcomes, reducing the prevalence of abdominal obesity is critical [7]. An increase in abdominal fat is directly linked to an imbalance between caloric intake and caloric expenditure. Therefore, the practice of activity is essential to increase caloric expenditure, improve quality of life and maintain a healthy weight, preventing the development of abdominal, and general, obesity [8].

Most of the population has not adhered to the guidelines of health organizations regarding the practice of regular physical activity [9]. The WHO (2020) recommends increasing levels of physical activity throughout the week for the entire population. At least 150–300 min or more of moderate physical activity, or at least 75–150 min or more of vigorous physical activity, or a combination of activities of moderate-vigorous intensity, throughout the week is recommended. Physical activity brings physical and mental health benefits [10]. In addition, it is recommended that the amount of time in sedentary behavior be reduced [11]. On the world stage, one in four adults and three in four adolescents do not follow physical activity recommendations [12]. Physical inactivity is associated with the development of chronic diseases, and some types of cancers, and, consequently, with a worsened quality of life [13].

Physical inactivity can accentuate the effects of menopause. Menopause is characterized by the reduction of estrogen and the suspension of menstruation. Among the effects that accompany this phase are an increase in body adiposity, especially in the central region, reduction of muscle mass, and other effects [14,15]. Menopause is associated with abdominal obesity; thus, in the analyses, it is of fundamental importance to observe women that are in menopause and those that are not in menopause.

A current behavior that has been considered a great problem for the health of the population is sedentary behavior, which is defined as any behavior, in a waking state, involving sitting, lying down or reclining positions with a caloric expenditure of 1.5 METs or less [16]. When practiced assiduously, sedentary behavior has emerged as a risk factor for several health problems, such as cardiovascular diseases, diabetes, and hypertension. [17,18]. Sedentary behavior is a type of behavior that can exist even when recommendations for physical activity are followed; that is, sedentary behavior is independent of physical activity levels [19]. The Physical Activity Guidelines Advisory Committee (PAGAC) reviewed the scientific evidence linking sedentary behavior to mortality, chronic disease, cardiovascular disease, cancer, and obesity, independent of physical activity levels [20]. Excess time in sedentary behavior increases the risk for developing several diseases, such as diabetes, hypertension and others. This type of behavior deserves further investigation, as much as does physical activity levels [21].

Decreased levels of physical activity and increased sedentary behavior are factors that lead to negative health outcomes, including abdominal obesity, which is a risk factor for cardiometabolic diseases. The study of this theme is of great importance for a better understanding of the association of these variables and, based on the results, to collaborate with health professionals for a better orientation of the population. Given the above, this study aimed to identify which combination of leisure-time physical activity and sedentary behavior most contributes to the prevention of abdominal obesity in adults participating in ELSA-Brasil (Longitudinal Study of Adult Health).

## 2. Materials and Methods

### 2.1. Population and Sample

The study design was cross-sectional and participants from the first follow-up (second wave) of the ELSA-Brasil cohort (2012–2014) were analyzed. ELSA-Brasil, Longitudinal Study of Adult Health, is the largest epidemiological and multicenter study in Latin America and aims to investigate the incidences of diabetes and cardiovascular diseases, and their risk factors, in the Brazilian population. It is a cohort, observational study that has been running for 15 years. The study involves the following six teaching and research institutions in different Brazilian cities: Salvador—Universidade Federal da Bahia (UFBA), Porto Alegre—Universidade Federal do Rio Grande do Sul (UFRGS), São Paulo—Universidade de São Paulo (USP), Rio de Janeiro—Fundação Oswaldo Cruz (Fiocruz), Belo Horizonte—Universidade Federal de Minas Gerais (UFMG), and Vitória—Universidade Federal do Espírito Santo (UFES). In this study, 15,105 civil servants from the six research institutions, from 35 to 74 years of age, were included in its baseline, and these civil servants answered questionnaires about their general health conditions, their family histories of diseases, mental health, use of medications, and physical activity, amongst other information [7,22,23].

ELSA-Brasil was approved by all the Research Ethics Committees (RECs) of the research centers involved and by the National Research Ethics Commission (CONEP). The Ethics Committee of the Instituto de Saúde Pública of the Universidade Federal da Bahia, under number 027–06/CEP-ISC, approved the study. All participants signed the free and informed consent form, with a guarantee of secrecy and confidentiality of data.

For the present study, all participants of the first follow-up (second wave—2012–2014) were selected, with the age range was between 38 and 78 years of age. Participants who did not answer the questionnaires on physical activity and sedentary behavior, and who did not collect data on abdominal obesity, were excluded, as were deaths and refusals.

### 2.2. Data Production

A team of interviewers and evaluators trained in the health field and certified by a quality control committee collected data. Training of the data collection team was performed centrally to ensure uniformity in the application of the study protocol in any ELSA-Brasil Research Center. Face-to-face interviews were conducted in the application of the questionnaire blocks [24]. The analysis was conducted between the months of May and June, 2023.

### 2.3. Abdominal Obesity Assessment

The dependent variable, abdominal obesity, was determined by means of waist circumference. The measurement was performed with the participant fasting and with an empty bladder. An anthropometric tape and a dermatographic pencil were used to mark the anatomical points. The standard measurement is made at the midpoint between the lower border of the costal arch and the iliac crest, on the median axillary line. If it is not possible to locate the anatomical points, this measurement can also be taken at the height of the umbilical scar. Abdominal obesity was classified, according to the International Diabetes Federation (IDF), as being >80 cm for women and >90 cm for men [7,25].

### 2.4. Assessment of Physical Activity

To identify and quantify physical activity, we used the long, validated Portuguese version of the International Physical Activity Questionnaire (IPAQ), which is composed of questions related to the frequency, intensity and duration of physical activity in four domains: work, commuting, household activities and free time [26]. In ELSA-Brasil, only the domains of physical activity during leisure time, and physical activity while commuting, were assessed. Physical activity was measured in minutes/week by multiplying the weekly frequency by the duration of each activity performed. For the purposes of this study, we used the leisure-time physical activity domain, which was categorized as 0 = insufficiently active (<150 min per week of moderate physical activity or walking and/or <60 min per week of vigorous physical activity and/or <150 min per week of any combination of walking, moderate or vigorous) and 1 = sufficiently active (>150 min per week of moderate physical activity or walking and/or >60 min per week of vigorous physical activity and/or >150 min per week of any combination of walking, moderate and vigorous). The cutoff points stipulated by IPAQ were used for physical activity [21,23,27].

### 2.5. Evaluation of Sedentary Behavior

Information on sedentary behavior was identified by interview, using a standard questionnaire applied in all ELSA-Brasil research centers, where participants answered questions about the numbers of hours spent sitting (cumulative sitting time) and the numbers of hours spent watching TV, playing video games, and using a cell phone or computer (leisure screen time) during a weekday and during a weekend day. A low level of sedentary behavior was classified as <2 h/day of screen time at leisure and <8 h/day of accumulated sitting time [21].

### 2.6. Assessment of Covariates

Covariates were collected using a standard questionnaire used in all ELSA-Brasil research centers. Age was categorized into three groups: age = 0, if between 38 and 50 years; age = 1, if between 51 and 60 years; and age = 2, if >60 years. For education, four strata were established: 0 = incomplete elementary; 1 = complete elementary; 2 = complete high school; and 3 = complete college/post-graduate. Current smoking status was categorized as no = 0 (non-smokers) and yes = 1 (smokers and former smokers). The variable excessive drinking was categorized as no = 0 and yes = 1. Excessive drinking was defined as >140 g (approximately 3500 mL p/week) for women and >210 g (approximately 5250 mL p/week) for men. Menopause was classified as no = 0 and yes = 1 [23,28,29].

### 2.7. Data Analysis

The statistical program STATA, version 14.0, was used. Descriptive measures were calculated for all categorized variables. The chi-square test was employed to analyze proportions. All analyses were stratified by sex a priori. The associations between the dependent variable (abdominal obesity) and the independent variables (different combinations of LPTA and sedentary behaviors) were analyzed by logistic regression. We estimated the OR (odds ratio) with 95 CI%. The following variables were considered as potential confounders or effect modifiers: age, education, excessive drinking, smoking, and menopause.

The criterion for the selection of effect-modifying variables was done through bivariate analysis, where stratum-specific point measures and their confidence intervals were analyzed. If the point measure of a factor in a specific stratum was not in the confidence interval of another factor in the same stratum, this would indicate an effect modification. From this analysis, none of the variables analyzed was an effect modifier for men, but for women, three variables proved to be possible effect modifiers: age, education, and menopause. To evaluate the potential effect modifier variables, a multivariate analysis was performed using the multiplicative model.

Confounding analysis was performed by comparing the OR for the crude association with that for the association adjusted for potential confounding factors. The parameter used to identify the difference between the two associations was 10%. Next, a logistic regression analysis was performed. The analysis started with a full model, followed by a one-by-one removal of each potential confounding variable.

Then, different logistic regression models were proposed using the following different combinations between LTPA and sedentary behaviors (cumulative sitting time and screen time at leisure during one day in the week and during one day on the weekend): insufficient LTPA and longer time in sedentary behavior (reference); sufficient LTPA and shorter time in sedentary behavior; insufficient LTPA and shorter time in sedentary behavior; sufficient LTPA and longer time in sedentary behavior. Each combination was analyzed separately for both men and women.

The final analysis model for men was adjusted for age and education, as the confounding variables observed in all combinations. The final model for women was as follows: the combination sufficient LTPA and less time in sedentary behaviors was stratified by age (0.0272) and menopause (0.0349) after multivariate analysis and adjusted for education. In this combination, women were divided into four groups: <50 years and >50 years/without menopause, <50 years and >50 years/with menopause. The combination of insufficient LTPA and too little time in sedentary behaviors was adjusted for age and education, and no effect-modifying variables were observed. The combination of sufficient LPTA and too much time in sedentary behaviors was stratified by menopause (0.0414), after multivariate analysis, and adjusted for age and education. In this combination, women were separated into two groups: those who were menopausal and those who were not menopausal. The confidence interval was set at 95%.

## 3. Results

A total of 14.014 participants were included in the analysis, of whom 6357 were men and 7657 women. A total of 1091 participants were excluded due to death (205) and refusals (886). The characteristics of the sample are presented in Table 1. Men had a higher household income than women (51.1%), smoked more (52.4%), consumed more beer (71.3%), spent more time sitting on a weekend day (53.3%), but were more physically active in their free time (51.2%). Women had higher levels of education, were more physically inactive (58.9%), spent more screen time in leisure time on both a weekday and a weekend day, values being 52.7% and 52%, respectively, and had a higher percentage of abdominal obesity (67%). There was a higher presence of menopausal women (85.3%). In this analysis, age and the amount of time spent sitting on a weekday were not statistically significant in the comparison between men and women.

In Table 2, the bivariate analysis of the association between leisure-time physical activity and sedentary behavior on abdominal obesity in men and women is presented. In this analysis, potential effect modifiers were observed only among women and the variables were age, education, and menopause. The variables were evaluated using multivariate logistic regression for each combination analyzed, as mentioned in the methodology.

The association between combinations of leisure-time physical activity (LTPA) and sedentary behaviors, accumulated sitting time and screen time in leisure time on a weekday and a weekend day, in abdominal obesity for men is presented in Table 3. All associations between the combinations of LTPA and SB were significant, but the combination was stronger when men were more physically active and spent less accumulated sitting time on sedentary behaviors on both a weekday and weekend day. The association between LTPA and sedentary behaviors appeared to reduce the chances of men developing abdominal obesity.

Table 4 shows the associations between the combinations of sufficient LTPA and little time in SB and sufficient LTPA and much time in SB among women. In the combination of sufficient LTPA and little time in a SB, either on a weekday or on a weekend day, it was observed that, among younger women who were not in menopause, most of the results were not statistically significant; in other words, even though these women were physically active and spent less time in a SB, the association did not seem to reduce the chances of developing abdominal obesity. Among older, non-menopausal women, the majority of the results showed that these women were more likely not to develop abdominal obesity. In women who were menopausal, both younger and older, the association seemed to reduce their chances of developing abdominal obesity. In the combination of sufficient LTPA and longer time in the sedentary behaviors analyzed, the results for the non-menopausal women were mostly significant. In the menopausal women, all results were statistically significant. Analyzing the two combinations, it was possible to observe that the menopausal women seemed to have a higher chance of not developing abdominal obesity when compared to the non-menopausal women, especially the younger ones.

In Table 5, the results of the combination of insufficient LTPA and little time in sedentary behaviors on both a day in the week and a day on the weekend on abdominal obesity in women were presented. All results were statistically significant; in other words, this combination appeared to protect women from developing abdominal obesity.

## 4. Discussion

In this study, we sought to investigate which combination of leisure-time physical activity and sedentary behavior contributed most to the prevention of abdominal obesity in both sexes.

It was observed that, among men, all combinations between LTPA and sedentary behaviors, accumulated sitting time and screen time during leisure time, showed statistically significant associations. However, the association was stronger when more physical activity was performed in leisure time and less time was spent in sedentary behaviors, during one day in the week and during one day on the weekend. It was also observed that even though more time was spent in sedentary behaviors and getting enough physical activity, the chances of developing abdominal obesity were reduced.

In one study, conducted in Chile, the joint effect of leisure-time physical activity and sedentary behavior on markers of adiposity and vascular risk was analyzed. The results showed that individuals who spent more time in sedentary behavior and performed more physical activity and those who spent less time in sedentary behavior and performed more physical activity were less likely to develop metabolic syndrome, have obese BMI, central obesity, diabetes, and hypertension [30]. In the Chilean study, among the adjustment variables analyzed in the models were age and education, and there were no effect modifier variables in the analysis of men and women. In the present study, effect modifiers were only observed in the analysis of women.

In the women, it was observed that, in the combination of sufficient LTPA and little time in SB, for the younger women, and those who were not in menopause, most of the results were not statistically significant, but among the older women the results seemed to reduce the chances of these women developing abdominal obesity. In the menopausal women, both younger and older, all results were significant; in other words, this combination seemed to protect these women from developing abdominal obesity. This combination was stratified by age and menopause and adjusted for education. The combination of sufficient LTPA and longer time in sedentary behaviors, was stratified by menopause and adjusted for age and education. In this combination, most of the results seemed to protect both menopausal and non-menopausal women from developing abdominal obesity.

In the studies found in the literature on the topic [30,31], the menopausal variable was not analyzed. Menopause is a phase in which there are many hormonal changes, increase in the percentage of fat in the abdominal region, reduction of muscle mass, and a greater probability of developing cardiometabolic diseases and cancer [14,15], making it an important variable for the analysis. Education was used as an adjustment variable in the previously mentioned studies, as well as in the present study. Education is associated with higher levels of physical activity in women over 60 years of age [32]. Which may explain the protection of physical activity in older women who were and were not in menopause.

In younger women who were not menopausal, the fact that the results were not statistically significant, even though these women were sufficiently active and spending less time in sedentary behaviors, may be explained by the low volume (time) of physical activity. In a study conducted in China [31], the women analyzed in the cohort showed a reduction in waist circumference when they performed more than 42 h/week of moderate to vigorous intensity physical activity when compared to women who spent less time on activity at these intensities.

Something that must be taken into account is the fact that women have less time to perform leisure-time physical activities. According to the study by Salles-Costa et al. [33], one of the justifications would be the multiple work shifts of women, which reduces the hours to perform LTPA. Aquino et al. [34] observed in their study, performed with nursing professionals, that half of the interviewees added another 20 h of domestic activities and child care to their intense work days during the week.

It is worth noting that the present study analyzed only one domain of physical activity, leisure-time physical activity. In the study by Hallal et al. [35], it was observed that there were no differences regarding the practice of physical activity between men and women when the domain of domestic activity, performed predominantly by women, was added. Therefore, domestic activities seem to be important for studies that aim to compare or quantify levels of physical activity between genders.

Regarding women in menopause, some studies show that increased levels of physical activities of moderate to vigorous intensity seem to prevent excess body fat in postmenopausal women [36,37]. Another important factor is that the climacteric phase and menopause are important phases for incorporating new habits, being times in which women can evaluate themselves [29,38].

The combination of insufficient LTPA and little time in sedentary behaviors was adjusted for age and education and for the potential effect modifier variables, none of which seemed to interfere with this combination. The results among women were similar to those among men for this combination. Women appeared to be less likely to develop abdominal obesity. In the study conducted in Chile, similar results were found for this combination among women [30].

According to the most recent WHO recommendations, increased levels of physical activity are needed to improve quality of life and health [11], and, for greater benefits, decreased time in sedentary behavior reduces the deleterious effects caused by sedentary behavior, such as cardiovascular disease, type 2 diabetes, cancer, obesity, and abdominal obesity, among others [11,21,39,40,41]. Even in combinations where sufficient LTPA and a long time in sedentary behaviors seemed to reduce the chances of developing abdominal obesity, it is worth noting that higher levels of physical activity can attenuate the deleterious effects of sedentary behavior, but not eliminate them [42].

Excessive time spent in sedentary behavior is associated with reduced LPL (lipoprotein lipase) activity. The enzyme LPL has the function of regulating the production of HDLs (high density lipoproteins) and the absorption of plasma triglycerides and is activated when there is muscle contraction; that is, when there is movement [21,43,44]. With a reduction in LPL activity, there is an increase of fat in the vessels. This fat falls into the bloodstream, reaching the central adipose tissue. This excess fat can lead to increased insulin resistance, causing hyperlipidemia, glucose intolerance and hypertension, and, ultimately, atherosclerosis [45,46].

The study has some limitations that need to be mentioned. Sedentary behavior does not have a defined cut-off point in the literature, so cut-off points already analyzed in other studies were used [21]. The collection of physical activity and sedentary behavior data was done through self-reported questionnaires, which might have led to errors and biases. Another limitation is that the population of ELSA-Brasil is not representative of the Brazilian population, since it is a sample of public servants, most of whom have a high level of education and income; however, the sample has a considerable size and presents regional and social diversity, as it derives from three important regions in Brazil, namely, the northeast, southeast, and south. Interviewers who were duly trained and certified by a quality control team carried out data collection. The instruments applied were all validated and standardized.

The results of this study can contribute to assisting public health managers and health professionals in guiding the population in regard to the importance of staying physically active, as well as to reducing time spent in sedentary behavior; thus, preventing negative health outcomes. Finally, further studies on this theme are recommended, analyzing other variables that may interfere in this association. Another interesting point would be the measurement of physical activity through accelerometry, a more direct measure, which would reduce memory biases.

## 5. Conclusions

The results showed that being sufficiently active and reducing time in sedentary behaviors was the combination that contributed most to the prevention of abdominal obesity in both men, older women who were not menopausal, and women who were menopausal.

In younger women who were not menopausal, the combination of being more physically active and spending less time in sedentary behaviors did not appear to reduce the odds of these women developing abdominal obesity. Thus, we see the need for further investigations regarding the variables that may influence this association in younger women, such as time and space available for physical activity, among others.

The results of the study reiterate the current recommendations proposed by health organizations that it is not enough just to increase levels of physical activity. It is also of paramount importance to reduce the time in sedentary behavior, which contributes to the improvement of public health.

## Figures and Tables

**Table 1 ijerph-20-06501-t001:** Characteristics of the sample at Wave 2. ELSA-Brasil (2012–2014).

	Men	Women	*p*-Value
AGE (years)	*n* (%)	*n* (%)	
38–50	2060 (46.71)	2350 (53.29)	
51–60	2193 (44.26)	2762 (55.74)	0.06
>60	2104 (45.26)	2545 (54.74)	
EDUCATION			
Incomplete Elementary	452 (62.78)	268 (37.22)	
Complete Elementary	488 (55.58)	390 (44.42)	**0.00**
Complete High School	1941 (44.14)	2456 (55.86)	
Complete college/Post-graduate	3467 (43.30)	4540 (56.70)	
SMOKING			
No	3297 (40.27)	4890 (59.73)	**0.00**
Yes	3049 (52.48)	2761 (47.52)	
EXCESSIVE DRINKING			
No	5529 (43.14)	7288 (56.86)	**0.00**
Yes	801 (71.39)	321 (28.61)	
MENOPAUSE			
No	-	1113 (14.64)	-
Yes	-	6492 (85.36)	
LEISURE-TIME PHYSICAL ACTIVITY			
Insufficiently active	3307 (41.09)	4742 (58.91)	**0.00**
Sufficiently active	3017 (51.29)	2865 (48.71)	
ABDOMINAL OBESITY			
No	4294 (55.16)	3490 (44.84)	**0.00**
Yes	2036 (33.0)	4133 (67.0)	
SEDENTARY BEHAVIOR			
Accumulated Sitting Time Weekday			
>8 h	1340 (44.94)	1642 (55.06)	0.59
<8 h	5017 (45.48)	6015 (54.52)	
Accumulated sitting time Weekday			
>8 h	737 (53.33)	645 (46.67)	**0.00**
<8 h	5620 (44.49)	7012 (55.51)	
Screen time at leisure Weekday			
>2 h	2355 (47.27)	2627 (52.73)	**0.00**
<2 h	4002 (44.31)	5030 (55.69)	
Screen time at leisure Weekday			
>2 h	3669 (47.92)	3987 (52.08)	**0.00**
<2 h	2688 (42.28)	3670 (57.72)	

Note: Values for men and women were compared using the chi-square test. The sum of the strata is not always equal due to the loss of information of some variables.

**Table 2 ijerph-20-06501-t002:** Crude and adjusted Odds Ratio (OR) of the association between leisure-time physical activity and sedentary behavior in abdominal obesity in participants of ELSA-Brasil.

	Men		Women	
Variables	OR	95 CI%	OR	95 CI%
Crude OR	0.73	(0.66–0.80)	0.89	(0.84–0.96)
**Age**				
38–50	0.73	(0.61–0.88)	**0.76**	**(0.68–0.84)**
>51	0.70	(0.63–0.78)	**0.91**	**(0.86–0.97)**
Adjusted OR	0.71	(0.65–0.78)	0.86	(0.81–0.91)
**Education**				
Up to high school complete	0.70	(0.51–0.96)	**1.04**	**(0.82–1.31)**
Incomplete college or higher	0.73	(0.66–0.80)	**0.86**	**(0.82–0.91)**
Adjusted OR	0.73	(0.66–0.80)	0.87	(0.82–0.92)
**Excessive drinking**				
No	0.72	(0.65–0.80)	0.88	(0.84–0.93)
Yes	0.75	(0.60–0.95)	0.91	(0.73–1.13)
Adjusted OR	0.73	(0.66–0.80)	0.89	(0.84–0.9)
**Smoking**				
No	0.71	(0.62–0.83)	0.90	(0.84–0.97)
Yes	0.73	(0.67–0.84)	0.87	(0.81–0.94)
Adjusted OR	0.73	(0.67–0.80)	0.89	(0.84–0.94)
**Menopause**				
No	-	-	**0.79**	**(0.70–0.89)**
Yes	-	-	**0.91**	**(0.85–0.96)**
Adjusted OR	-	-	0.88	(0.84–0.93)

Note: OR: Odds Ratio; CI: Confidence Interval.

**Table 3 ijerph-20-06501-t003:** Association between combinations of leisure-time physical activity and sedentary behavior (cumulative sitting time and screen time at leisure) on one day in the week and one day in the weekend on abdominal obesity in male participants of ELSA-Brasil. ELSA-Brasil (2012–2014).

Combinations between Physical Activity and Sedentary Behavior	Men (95 CI%)
DAY OF THE WEEK	
Insufficient LTPA—Long cumulative sitting time	**1.00 (Reference)**
Insufficient LTPA—Short accumulated sitting time	0.68 (0.57–0.81)
Sufficient LTPA—Long cumulative sitting time	0.61 (0.48–0.77)
Sufficient LTPA—Short accumulated sitting time	**0.51 (0.43–0.60)**
WEEKEND	
Insufficient LTPA—Long cumulative sitting time	**1.00 (Reference)**
Insufficient LTPA—Short accumulated sitting time	0.65 (0.53–0.80)
Sufficient LTPA—Long cumulative sitting time	0.72 (0.53–0.98)
Sufficient LTPA—Short accumulated sitting time	**0.47 (0.38–0.58)**
DAY OF THE WEEK	
Insufficient LTPA—Long leisure screen time	**1.00 (Reference)**
Insufficient LTPA—Short leisure screen time	0.67 (0.58–0.77)
Sufficient LTPA—Long leisure screen time	0.74 (0.62–0.88)
Sufficient LTPA—Short leisure screen time	**0.49 (0.42–0.58)**
WEEKEND	
Insufficient LTPA—Long leisure screen time	**1.00 (Reference)**
Insufficient LTPA—Short leisure screen time	0.69 (0.60–0.80)
Sufficient LTPA—Long leisure screen time	0.74 (0.64–0.85)
Sufficient LTPA—Short leisure screen time	**0.49 (0.42–0.58)**

Note: Insufficient LTPA (<150 min per week of moderate physical activity or walking and/or <60 min per week of vigorous physical activity and/or <150 min per week of any combination of walking, moderate or vigorous); Values adjusted for age and education.

**Table 4 ijerph-20-06501-t004:** Association between the combinations of sufficient LTPA and little time in sedentary behaviors and sufficient LPTA and much time in sedentary behaviors, both on a day in the week and on a day in the weekend in abdominal obesity in women participating in ELSA-Brasil (2012–2014).

Combinations between Physical Activity and Sedentary Behavior	No Menopause		Menopause	
	≤50 Years	>50 Years	≤50 Years	>50 Years
Insufficient LTPA—Long time spent in SB	1.00 (Ref.)	1.00 (Ref.)	1.00 (Ref.)	1.00 (Ref.)
Sufficient LTPA and less accumulated sitting time on a weekday ^a^	**0.46 (0.24–0.88)**	**0.37 (0.24–0.60)**	**0.36 (0.27–0.48)**	**0.63 (0.51–0.78)**
Sufficient LTPA and less accumulated sitting time on a weekend day ^a^	0.63 (0.22–1.79)	0.53 (0.27–1.06)	**0.24 (0.16–0.37)**	**0.58 (0.43–0.78)**
Sufficient LTPA and less screen time at leisure on a weekday ^a^	0.57 (0.29–1.10)	**0.37 (0.24–0.58)**	**0.34 (0.26–0.46)**	**0.47 (0.39–0.56)**
Sufficient LTPA and less screen time at leisure on a weekend day ^a^	0.53 (0.27–1.03)	**0.40 (0.26–0.62)**	**0.37 (0.27–0.49)**	**0.57 (0.48–0.68)**
	**No Menopause**		**Menopause**	
Insufficient LTPA—Long time spent in SB	1.00 (Ref.)		1.00 (Ref.)	
Sufficient LTPA and more accumulated sitting time on a weekday ^b^	**0.47 (0.28–0.79)**		**0.61 (0.48–0.77)**	
Sufficient LTPA and more accumulated sitting time on a weekend day ^b^	0.64 (0.28–1.46)		**0.42 (0.29–0.63)**	
Sufficient LTPA and more screen time at leisure on a weekday ^b^	**0.57 (0.36–0.89)**		**0.70 (0.58–0.84)**	
Sufficient LTPA and more screen time at leisure on a weekend day ^b^	**0.59 (0.42–0.84)**		**0.64 (0.55–0.74)**	

Note: Sufficient LTPA (>150 min per week of moderate physical activity or walking and/or >60 min per week of vigorous physical activity and/or >150 min per week of any combination of walking, moderate or vigorous); ^a^—values stratified by age and menopause and adjusted for education; ^b^—values stratified by menopause and adjusted for age and education; Ref.—Reference.

**Table 5 ijerph-20-06501-t005:** Association between the combination of insufficient LTPA and little time in sedentary behaviors, both on a day in the week and on a day in the weekend, in abdominal obesity in women participating in ELSA-Brasil (2012–2014).

Combination between Physical Activity and Sedentary Behavior	
Insufficient LTPA—Long time spent in SB	**1.00 (Ref.)**
Insufficient LTPA and less accumulated sitting time on a weekday	**0.80 (0.70–0.93)**
Insufficient LTPA and less accumulated sitting time on a weekend day	**0.70 (0.57–0.88)**
Insufficient LTPA and less screen time at leisure on a weekday	**0.72 (0.63–0.81)**
Insufficient LTPA and less screen time at leisure on a weekend day	**0.80 (0.71–0.90)**

Note: Insufficient LTPA (<150 min per week of moderate physical activity or walking and/or <60 min per week of vigorous physical activity and/or <150 min per week of any combination of walking, moderate or vigorous); Values adjusted for age and education; Ref.—Reference.

## Data Availability

Not applicable.

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
