# Peer review of "Combined Effect of Leisure-Time Physical Activity and Sedentary Behavior on Abdominal Obesity in ELSA-Brasil Participants"

_ijerph, 2023, doi:10.3390/ijerph20156501_

Round 1
Reviewer 1 Report
I am very happy that the author was able to write such a wonderful paper and I congratulate the author on his achievement. This paper is relatively standardized and the data analysis part is well organized. However, I think the paper has the following points for improvement.
First, in the introduction section, it is recommended not to divide it into too many paragraphs. For example, the first few paragraphs of the introduction talk about a single issue and it is recommended to merge them appropriately.
Second, another part of the introduction is the literature review, which is reflected in the later paragraphs of the introduction. However, the authors currently do not highlight what kind of highlights their research and previous studies have. I suggest adding slightly more recent literature and summarizing and concluding it.
Third, for the P-values in Table 1, the decimal points should be standardized and in blue font unnecessarily.
Fourth, the paper should conclude with the corresponding contribution of the study, which is currently not explored enough in this manuscript.
Fifth, in Figure 1, the paper proposes the mechanism, but you don't actually test it right? I think this part can be weakened appropriately.
Sixth, for the participating literature section, please use English uniformly, such as the second reference.
Seventh, I noticed that in the references, there is no recent research, please consider adding literature from 2022 and 2023 appropriately to highlight the cutting edge of the research. For example, DOI: 10.3390/ijerph191912383
Reviewer 2 Report
Dear Authors.
Congratulation on work well done, and thank you for the opportunity to review this great work. The manuscript is well written in general. However, I would like to make some minor suggestions to enhance the ease of your prospective readers in the future.
Suggestions:
1. ELSA-Brazil: may I suggest providing some background on what “ELZA” represent and what it entails? This will benefit those, such as myself, who have little understanding of ‘ELZA-Brazil beside assuming it is an exercise programme.
2. Introduction: please try to avoid single-sentence paragraphs.
3. Line 39: “Between 1975 and 2016”, please provide some reference or data for the 1975 situation. The current statistics are mainly for 2016 or 2002 to 2019.
4. Line 73: “the development of numerous diseases” , to add “such as XXX (example of diseases).
5. Data Production: I suggest indicating the period of data analysis since it is already indicated the analysis is done based on the cohort of participants from 2012 to 2014.
6. Assessment of physical activity: Line 133, IPAQ, can I assume the Brazilian Portuguese version was used? Which is “comprised of”, a typo error?
7. Tables 2 and 3. May I suggest to standardise the abbreviation of 95% confidence interval to 95% CI? There are at least 2 versions used in the tables.
8. Table 3: “Referencial”: please consider using the English word equivalent.
Some simple cross check of English typo error or the use of non-English words is necessary.
Reviewer 3 Report
The authors conducted a cross-sectional study to assess the relationship between physical activity and obesity. I'm impressed with the size of the sample, I think this is one of the 'fortes' of this study. Obrigado to be able to review this manuscript.
Authors should do a minor editing before the manuscript could be considered for publication, even though I believe it is really close to the final stage.
ABSTRACT
line 17. This is more like "Objective", there is no real "Introduction" here
line 18. Please try not to use abbreviations in the Abstract
line 19. What is ELSA-Brazil? I recommend you should present it before mentioning it
line 21. Could you use the term "outcomes"? And provide the name of the tool used to assess physical activity and sedentary behavior. This will be more precise
line22. Same. I think it is more interesting to name the instrument you used to assess abdominal obesity
line 33. ELSA-Brazil. This is not an appropriate keyword, I recommend to select keywords contained within a scientific thesaurus
INTRODUCTION
line 90. Many readers, as myself, could be unaware of what ELSA-Brazil is, so it it interesting to present the meaning of it in the Introduction as well. Please add.
METHODS
lines 111-113. You should not provide the information about the sample here, this should be given in Results. You must mention the elegibility criteria here, instead.
line 118. Certified interviewers and scorers. What does this mean? What was their professional experience? Journalism?
lines 132-135. Did you use a validated protuguese version of the questionnaire? This is an important issue to ensure construct validity, I'm not sure if you were directly translating the English version or using a validated translated version
line 169-171. I suggest they should all be mentioned in the Abstract, then. It seem like if you have omited some of the outcomes, maybe due to convenience
line 203. STATA. This should be mentioned at the beggining of this section
RESULTS
line 206. "are shown", don't use past tense here
TABLE 1
Don't write the sample size here
You should add what I assume is the SD to clarify what the number in brackets after the results means
TABLE 3
Don't use Portuguese, be consistent with English. (Referência)
TABLE 4
You should state in the legend what "Ref." means, even though it is obvious
DISCUSSION
I'm not sure if the Figure is really necessary
CONCLUSIONS
Conclusions are way too long, you should be more concise. Conclusions must match the objectives of the study, they should be clear and concise
REFERENCES
Please check the references
Some of them are written in Portuguese
Some are written in a incorrect format
Please adhere to the recommendations and unify reference styles
You should use a reference manager to ensure this.
